# Effects of Perioperative Oral Management in Patients with Cancer

**DOI:** 10.3390/jcm11216576

**Published:** 2022-11-06

**Authors:** Yasuhiro Kurasawa, Akihiko Iida, Kaya Narimatsu, Hideki Sekiya, Yutaka Maruoka, Yukihiro Michiwaki

**Affiliations:** 1Maxillofacial Surgery, Graduate School, Tokyo Medical and Dental University, Tokyo 113-8520, Japan; 2Department of Dentistry and Oral Surgery, Nagaoka Red Cross Hospital, Niigata 940-2085, Japan; 3Department of Oral Surgery, School of Medicine, Toho University, Tokyo 143-8541, Japan; 4Oral and Maxillofacial Surgery, Center Hospital of the National Center for Global Health and Medicine, Tokyo 162-8655, Japan

**Keywords:** perioperative oral management, incidence of pneumonia, mortality

## Abstract

Perioperative oral management (POM) is used to prevent pneumonia in patients with cancer. However, the factors that expose hospitalized patients to increased risk of developing pneumonia remain unclear. For example, no study to date has compared the incidence of pneumonia in hospitalized patients by cancer primary lesion, or POM implementation, or not. We determined which patients were most likely to benefit from POM and examined the effects of POM on pneumonia prevention and mortality. In a total of 9441 patients with cancer who underwent surgery during hospitalization, there were 8208 patients in the No POM group, and 1233 in the POM group. We examined between-group differences in the incidence of pneumonia and associated outcomes during hospitalization. There was no significant between-group difference in the incidence of pneumonitis, however, patients with lung, or head and neck cancers, demonstrated a lower incidence of postoperative pneumonia. Among patients with lung and pancreatic cancers, mortality was significantly lower in the POM group. POM appears effective at reducing the risk of postoperative pneumonia in patients with certain cancers. Further, mortality was significantly lower in patients with lung and pancreatic cancers who received POM; hence, POM may be an effective adjuvant therapy for patients with cancer.

## 1. Introduction

Patients with cancer may receive perioperative oral management (POM) to help prevent the aspiration of oral pathogens during surgery (potentially leading to aspiration pneumonia), prevent surgical site infections, and prevent oral mucositis while undergoing adjuvant therapies such as radiotherapy and chemotherapy [1,2,3]. Of these, pneumonia is a major risk factor of poor progress for patients with cancer, with an incidence of 2–3% [4,5,6]. Aspiration pneumonia can occur during hospitalization [7], and it is particularly dangerous in patients with poor oral hygiene [1,2,8]. POM and enhanced oral hygiene may therefore help prevent perioperative pneumonia in patients with cancer.

In a multicenter study using the Diagnosis Procedure Combination (DPC) data, the authors reported that the incidence of pneumonia after hospital admission was lower two years before POM was listed in Japan’s social insurance coverage, than two years after insurance coverage, among 25,000 patients with cancer who underwent surgery [9]. While this study was well powered, and its results reliable, the authors compared the periods before and after POM was covered by Japan’s social insurance. There was no examination of the outcomes related to POM. Accordingly, we examined outcomes related to POM, including the incidence and outcome of pneumonia during hospitalization. To accomplish this, we identified approximately 10,000 patients with cancer who underwent surgery at a single institution from 2011 to 2018 to examine the potential effects of POM on pneumonia prevention and mortality.

## 2. Materials and Methods

### 2.1. Patients

There were 77,805 patients admitted to Nagaoka Red Cross Hospital in Japan from April 2011 to March 2018. We excluded 6579 patients who were diagnosed with pneumonia at the time of admission. In all, 9441 patients with cancer who underwent surgery while hospitalized were included. The patients were classified into two groups based on whether or not they received POM during hospitalization: 8208 patients did not receive POM (No POM group), while 1233 did (POM group) (Figure 1).

### 2.2. Variables

We used the DPC database to select patients. The details of the DPC database have been described elsewhere [9,10,11,12,13]. The DPC data of hospitalized patients included all electronic records pertaining to clinical and medical care information. We recorded each patient’s sex, age, reason for admission, admission date, discharge date, primary diagnosis, complications during hospitalization, and outcome at discharge. For patients in the POM group, we noted the start date of POM.

The DPC database includes a 14-digit code called the DPC code. This code is specific to each patient and describes the patient’s cause of hospitalization, whether surgery or treatment was performed, the presence of side effects, and others. Further, this code is associated with use of the most medical resources at the time of discharge. The primary diagnosis was determined by identifying the DPC code, and we identified all patients with cancer from the database using the top six digits indicating the type of cancer. Patients who did not undergo surgery and who carried a diagnosis of pneumonia at the time of hospitalization were excluded. The patient was considered to demonstrate post-hospitalization pneumonia if pneumonia did not trigger the original hospitalization, and was not listed among the patient’s comorbidities on admission.

POM is not a standardized procedure; the POM group consisted of patients undergoing cancer treatment who were referred to dentistry and started POM prior to discharge. The remaining surgical patients with cancer comprised the No POM group. We also calculated the percentage of patients who underwent POM for each site of origin (i.e., performing rate of POM: PRP). PRP was calculated based on the number of POM patients/total number of patients by primary lesion. Outcomes at discharge were classified as survival or death, and the percentage of patients who died at the time of discharge (i.e., mortality) was calculated for each primary lesion.

### 2.3. Statistical Analysis

The IBM SPSS Statistics 26 was used for the statistical analysis. The t-test, chi-square test, Fisher’s exact probability test, and logistic regression analysis were used for the statistical analysis. In the logistic regression analysis, the status of pneumonia during hospitalization was used as the objective variable. POM implementation, sex, age, and cancer type were used as the explanatory variables. A *p*-value of <0.05 indicated a significant difference.

## 3. Results

### 3.1. Overview of the Target

Of the 9441 patients, 8208 were in the No POM group and 1233 in the POM group. At baseline, the groups exhibited significant differences in age and sex. The average age was 65.0 ± 14.2 years in the No POM group and 66.9 ± 11.5 years in the POM group (*p* < 0.001). In the No POM group, 50.6% were male; in the POM group, 59.0% were male (*p* < 0.001) (Table 1).

In the No POM group, the most common primary cancers were breast (12.6%), followed by stomach (12.0%), hematopoietic (11.7%), gynecologic (11.3%), colorectal (10.4%), lung (9.1%), and bladder (8.4%). In the POM group, the most common cancers were colon (33.3%), stomach (24.6%), lung (14.2%), head and neck (5.9%), and liver and gallbladder (5.0%). In the POM group, only 1.1% had breast cancer as their primary lesion (Figure 2, Appendix A).

### 3.2. Incidence of Pneumonia by a Primary Lesion

In the breast, gynecologic, prostate, kidney, small intestine, skin, eye, and male genital cancers, no patient developed pneumonitis. The incidence of pneumonia was also low among those with liver and gallbladder cancers (0.7%), thyroid gland cancer (0.6%), and pancreatic cancer (0.4%).

Patients with primary lesions associated with a high incidence of pneumonia included patients with esophageal (5.2%), hematopoietic (4.8%), musculoskeletal (4.1%), lung (3.7%), head and neck (3.7%), brain (2.0%), colorectal (1.4%), and stomach cancers (1.3%) were among the most common. In the POM group, the incidence at these sites was as follows: esophageal cancer (5.0%), hematopoietic tumor (4.8%), musculoskeletal cancer (0%), lung cancer (1.1%), head and neck cancer (2.7%), brain tumor (0%), colorectal cancer (1.5%), and stomach cancer (1.0%) (Table 2).

### 3.3. Examination of the Effects of POM

We investigated the efficacy of POM for preventing pneumonia in patients with primary lesions associated with a higher incidence of pneumonia in the No POM and POM groups. While we found no statistically significant differences, there were reductions in the prevalence of pneumonia in patients with musculoskeletal (4.1%), lung (2.6%), brain (2.0%), head and neck (1.0%), thyroid gland (0.6%), pancreatic (0.4%), stomach (0.3%), and esophageal cancers (0.2%).

Mortality rates in the No POM group were higher among those with hematopoietic (15.1%), musculoskeletal (12.2%), pancreatic (8.7%), small intestine (5.1%), liver and gallbladder cancer (5.1%), brain (4.6%), lung (4.3%), and esophageal cancers (4.1%). In the POM group, mortality was greatest for patients with hematopoietic cancers (21.4%) and gallbladder cancer (1.6%). Death did not occur during the study period for patients with musculoskeletal, pancreatic, small intestine, liver, and brain cancers. Statistically significant differences between the POM and No POM groups were observed for patients with lung and pancreatic cancers, with significantly lower mortality in the POM group (Table 2).

### 3.4. Comparison of the Degree of Reduction in the Incidence of Pneumonia by Primary Lesion and the Proportion of Patients Who Underwent POM (PRP)

PRPs were highest in colon cancer (32.4%), followed by those in stomach cancer (23.6%), head and neck cancer (19.6%), lung cancer (18.9%), and pancreatic cancer (15.9%) (Table 3).

Subsequently, we compared the differences in the incidence of pneumonitis between the POM and No POM groups with PRP for the 11 primary lesions with at least 10 cases in the POM group (colon, stomach, lung, head and neck, liver and gallbladder, pancreatic, hematopoietic tumor, kidney, thyroid gland, esophageal, and breast cancers). Among these, the incidence of pneumonia in the POM group was lower than that in the No POM group for lung, head and neck, cancer, pancreatic, stomach, and esophageal cancers by 0.2–2.6%. Differences between the POM group and the No POM group were noted for kidney and hematopoietic cancers. In the POM group, the incidence of pneumonia was higher than in the No POM group for colon and gallbladder cancers, increasing from 0.1–0.9%. There was no significant correlation between PRP and changes in pneumonia prevalence (Figure 3).

### 3.5. Factor Analysis of Pneumonia Occurring during Hospitalization

Concerning the use of POM and the incidence of pneumonitis, the odds ratio of POM was 0.87 (*p* = 0.609) using No POM as the reference standard. There were no significant between-group differences.

Regarding sex, the odds ratio for males was 2.52 (*p* < 0.001).

Based on age under 50 years, the odds ratio increased with increase in age, and the risk of pneumonia increased significantly in people aged over 60 years (odds ratio 3.53, *p* = 0.017), 70 s (odds ratio 3.65, *p* = 0.014), and 80 s or older (odds ratio 5.69, *p* = 0.001) age groups.

Using sites with a pneumonia incidence of <1% as the reference standard, the odds ratio by primary lesion with a pneumonia incidence of ≥1% was highest for hematopoietic tumor at 15.80 (*p* < 0.001), followed by musculoskeletal cancer at 14.82 (*p* < 0.001), esophageal cancer at 13.11 (*p* < 0.001), and lung cancer at 10.32 (*p* < 0.001), indicating elevated risk (Table 4).

## 4. Discussion

Patients’ risk of postoperative complications varied by primary cancer. The incidence of pneumonia reduced for some cancers; however, there was no statistical difference in the incidence of pneumonia between the POM and No POM groups. Notably, the mortality rate was significantly lower in the POM group than in the No POM group.

### 4.1. Differences in the Incidence of Pneumonia According to the Primary Cancerous Lesion

Factors which can cause aspiration pneumonia to develop in the postoperative period include deterioration in the patient’s general condition after surgery, decreased respiratory and swallowing functions, and oral and pharyngeal contamination. In this study, patients with breast, gynecologic, prostate, and kidney cancers did not develop pneumonia. The incidence of pneumonia was highest in patients with esophageal (5.2%), hematopoietic (4.8%), musculoskeletal (4.1%), lung (3.7%), head and neck (3.7%), and brain cancers (2.0%).

Patients with hematopoietic tumors exhibited immunocompromise and activities of daily living were often reduced in patients with musculoskeletal tumors. Brain, esophageal, and head and neck cancers were generally associated with longer operative times and more blood loss. In patients with esophageal cancer, smoking and alcohol consumption were frequent and esophageal motility disorders, common in this population, often contributed to preoperative malnutrition [14,15]. Patients with lung cancer were also more likely to be smokers and had reduced respiratory function and clearance capacity [16]. The operative field is also continuous with the oral cavity and pharynx. However, in breast, gynecologic, and prostate cancers, the operative field is located further away from the airway.

Therefore, the incidence of pneumonia according to the primary cancer could also be influenced by the patient’s general preoperative condition and nutritional status, the degree of contamination in the oral cavity, and the relationship between the operative field and the airway.

Given these findings, POM appears especially important for use with patients diagnosed with esophageal, hematopoietic, musculoskeletal, lung, head and neck, and brain cancers. These cancers are associated with general, preoperative vulnerabilities, highly invasive procedures, and a surgical field that transverses the airway.

### 4.2. Efficacy of POM in Preventing Pneumonia

POM reduced the incidence of pneumonia in patients with musculoskeletal, lung, brain, head and neck, thyroid gland, pancreatic, stomach, and esophageal cancers. Except for thyroid gland cancer and pancreatic cancer, in which the incidence of pneumonia was relatively low, we confirm that POM is indicated for patients with cancers associated with a high incidence of pneumonia. However, even if the incidence of pneumonia as classified from the primary lesion was not high, POM implementation appeared effective in patients with immunocompromise, respiratory dysfunction, and oral and pharyngeal contamination.

### 4.3. Review of the Literature on Pneumonia Prevention Effect of POM

Kurasawa et al. reported that 2% of patients with cancer develop pneumonia after hospitalization [9]. The incidence of pneumonia after hospitalization in this study was similar at 1.6%. When examining the efficacy of POM in preventing pneumonia by cancer type, POM helps prevent postoperative pneumonia in patients with esophageal, lung, and other cancers [17,18,19]. This study showed a reduction in the incidence of pneumonia by approximately 2.6% in patients with lung cancer and 1.0% in patients with head and neck cancer. This suggests that POM effectively inhibits the development of pneumonia in patients with certain cancers. However, the differences were not statistically significant.

We believe that there were two reasons for the non-significant between-group differences. First, we did not adjust for factors involved in the development of pneumonia. The study cohort consisted of patients with different cancers and at different stages of cancer progression. The patients also underwent different surgeries, had different comorbidities, and had varying oral cavity contamination, even among those with the same primary cancer. Unfortunately, in Japan, it is difficult to fully control these factors because POM can be implemented at the request of the primary medical department. Ishimaru et al. reported a 0–48% reduction in the risk of postoperative pneumonitis in patients undergoing cancer surgery who received preoperative oral care by a dentist [20]. However, Sekiya et al. noted that Ishimaru et al.’s study did not adjust for oral hygiene, which may have affected the results of their analysis. Sekiya et al. established an “oral triage system” that selects patients who need POM. The system confirmed each patient’s oral hygiene status before surgery under general anesthesia and compared the incidence of pneumonia in patients with cancer before and after the surgery. Use of the oral triage system reduced the risk of pneumonia after surgery [21]. Poor oral hygiene status may be a criterion for POM since oral contamination is a risk factor for aspiration pneumonia. This study included patients with various oral hygiene conditions; POM was probably not necessary for some and this may have influenced the statistical analysis results.

Second, there was bias regarding the site of origin where POM was performed. Three primary lesions in the POM group were in the stomach, lung, and colon, accounting for 70% of the total sites. PRP in these primary lesions was 23.6% in stomach cancer, 18.9% in lung cancer, and 32.4% in colon cancer. The incidence of postoperative pneumonia in cases with these primary lesions was reportedly 2.2–4.3% [22,23] for stomach cancer, 1.2–5.6% [24,25] for lung cancer, and 1.8% [26] for colon cancer, which was similar to that in this study. During the multivariate analysis, the odds of developing pneumonia was 3.34 for stomach cancer, 4.43 for colon cancer, and 10.32 for lung cancer. The risk of developing pneumonia in patients with stomach cancer and colon cancer was relatively low. However, hematopoietic (15.8), esophageal (13.1), and head and neck cancers (9.76) were associated with higher risk; thus, for preventing pneumonia, medical resources should be directed to primary lesions associated with higher risk.

Comparing the POM and No POM groups, statistically significant differences were observed in mortality. When we analyzed by the type of cancer, significant differences were observed for lung and pancreatic cancers. Ishimaru et al. examined 509,179 patients who underwent cancer surgery. They found that preoperative oral care, by dentists, was associated with a significant reduction in all-cause 30-day postoperative mortality [20]. The mechanisms that affect the prognosis of patients were not clearly elucidated in this study. However, improved oral hygiene and swallowing function presumably reduce aspiration pneumonia [1,27] and surgical site infections [28]. These results suggest that POM can be an effective supportive treatment for patients with cancer.

Although it was difficult to perform statistical analysis by adjusting for the patients’ general conditions in this study, details of cancer treatment, cause of death, oral hygiene, POM content, and so on, it was necessary to appropriately arrange medical resources considering the incidence of pneumonia, mortality rate, and the risk of pneumonia in cases with each of the primary lesion sites as identified in this study.

### 4.4. Effect of POM in Reducing Mortality

Since postoperative pneumonia may be fatal, reducing the incidence of postoperative pneumonia may also reduce postoperative mortality. In this study, the mortality rates were significantly lower in the POM group with lung (*p* = 0.018) and pancreatic cancers (*p* = 0.019). Since the analysis of individual cases was difficult in this study, it was not possible to clarify the role POM played in reducing mortality. This potential relationship should be examined in future studies.

## 5. Conclusions

For patients with certain types of cancer, such as lung, and head and neck cancers, POM appears to reduce the onset of pneumonia after hospitalization. Because POM was frequently performed in patients with colon and stomach cancers, who had a relatively low risk of developing pneumonia, the redistribution of medical resources to higher risk patients might help prevent pneumonia and its associated mortality in this population. Mortality was significantly lower in patients who received POM; thus, POM contributed to prolonging the prognosis of patients. POM may be effective as adjuvant therapy for patients with cancer.

## Figures and Tables

**Figure 1 jcm-11-06576-f001:**
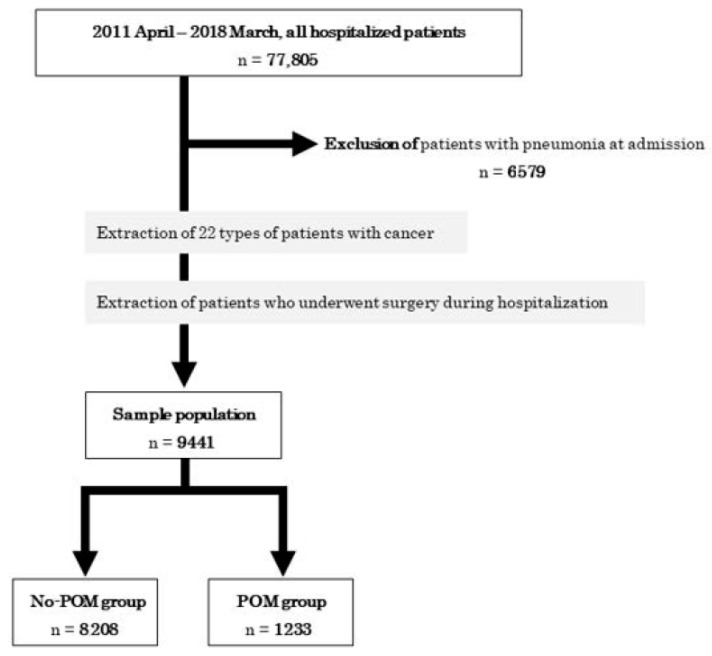
Scheme of patient selection.

**Figure 2 jcm-11-06576-f002:**
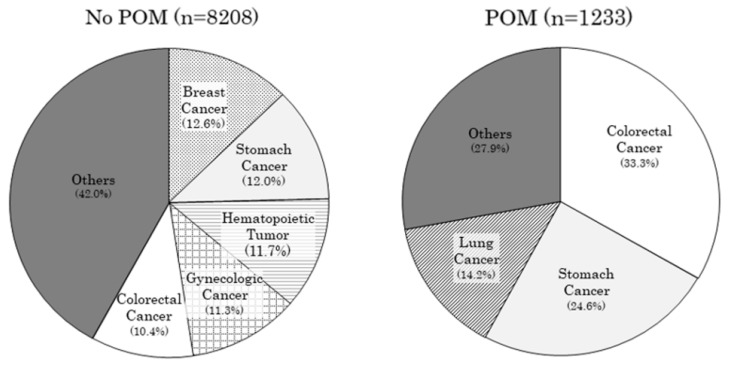
Comparison of the percentage of primary lesions in No POM and POM.

**Figure 3 jcm-11-06576-f003:**
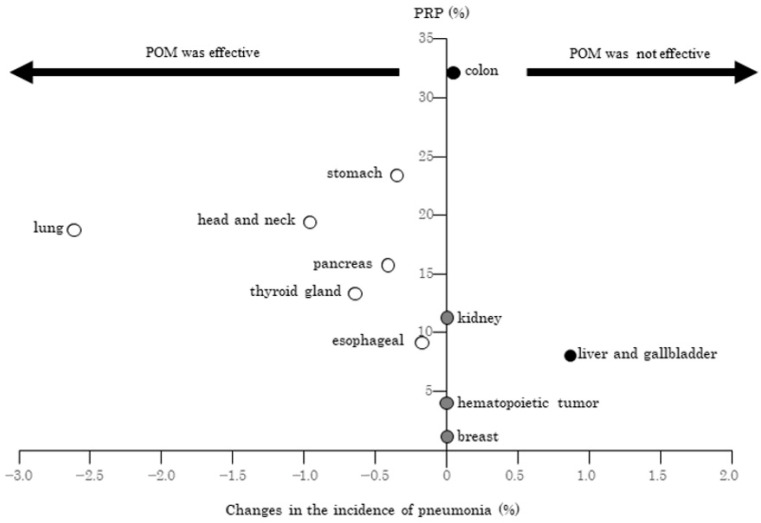
Relationship between changes in the incidence of pneumonia and performing rate of POM.

**Table 1 jcm-11-06576-t001:** Characteristics of the No POM and POM groups.

	No POM		POM		
	(n = 8208)		(n = 1233)		*p*-Value
Age (year), mean ± SD	65.0	14.2	66.9	11.5	<0.001
Sex (male), n (%)	4153	50.6	728	59.0	<0.001

**Table 2 jcm-11-06576-t002:** Comparison of pneumonia incidence and mortality by the primary lesion.

	Incidence of Pneumonia (%)				Mortality (%)			
Primary Lesion	No POM	POM	Difference No POM and POM	*p*-Value	No POM	POM	Difference No POM and POM	*p*-Value
Esophageal cancer	5.2	5.0	−0.2	NS	4.1	0	−4.1	NS
Hematopoietic tumor	4.8	4.8	0	NS	15.1	21.4	6.3	NS
Musculoskeletal cancer	4.1	0	−4.1	NS	12.2	0	−12.2	NS
Lung cancer	3.7	1.1	−2.6	NS	4.3	0.6	−3.7	0.018
Head and neck cancer	3.7	2.7	−1.0	NS	1.3	1.4	0.1	NS
Unknown primary	3.2	100	96.8	NS	12.9	0	−12.9	NS
Brain tumor	2.0	0	−2.0	NS	4.6	0	−4.6	NS
Colorectal cancer	1.4	1.5	0.1	NS	2.0	2.0	0.0	NS
Stomach cancer	1.3	1.0	−0.3	NS	3.9	2.6	−1.3	NS
Liver and gallbladder cancer	0.7	1.6	0.9	NS	5.1	1.6	−3.5	NS
Thyroid gland cancer	0.6	0	−0.6	NS	1.3	0	−1.3	NS
Pancreas cancer	0.4	0	−0.4	NS	8.7	0	−8.7	0.019
Breast cancer	0	0	0		1.0	0	−1.0	
Gynecologic cancer	0	0	0		2.4	0	−2.4	
Prostate cancer	0	0	0		1.9	0	−1.9	
Kidney cancer	0	0	0		3.2	0	−3.2	
Small intestine cancer	0	0	0		5.1	0	−5.1	
Mediastinal cancer	0	0	0		0	0	0	
Skin cancer	0	0	0		0			
Malignant eye tumor	0				0			
Male genital cancer	0				10.0			
Malignant cardiac tumor	0				25.0			

NS: not significant.

**Table 3 jcm-11-06576-t003:** Comparison of PRP by the primary lesion.

Primary Lesion	No POM (n)	POM (n)	PRP (%)
Colorectal cancer	857	410	32.4
Stomach cancer	982	303	23.6
Head and neck cancer	299	73	19.6
Lung cancer	749	175	18.9
Pancreas cancer	253	48	15.9
Thyroid gland cancer	160	25	13.5
Kidney cancer	279	32	11.5
Esophageal cancer	194	20	9.3
Liver and gallbladder cancer	693	62	8.2
Hematopoietic tumor	958	42	4.2
Breast cancer	1033	14	1.3
Small intestine cancer	59	7	10.6
Mediastinal cancer	54	6	10.0
Prostate cancer	376	6	1.6
Musculoskeletal cancer	74	1	1.4
Brain tumor	153	2	1.3
Gynecologic cancer	930	6	0.6
Malignant eye tumor	20	0	0
Male genital cancer	10	0	0
Malignant cardiac tumor	4	0	0
Skin cancer	40	0	0
Other	31	1	3.1

**Table 4 jcm-11-06576-t004:** Risk factors for pneumonia during hospitalization.

	n	Odds Ratio	95% CI		*p*-Value
Female	4560	standard			
Male	4881	2.52	1.67	3.80	*p* < 0.001
Age <50 years	1328	standard			
50–59	1321	1.83	0.57	5.89	NS
60–69	2694	3.53	1.25	9.94	0.017
70–79	2853	3.65	1.30	10.22	0.014
80 or older	1245	5.69	1.99	16.26	0.001
No POM	8208	standard			
POM	1233	0.87	0.52	1.47	NS
Others	4149	standard			
Brain tumor	155	9.33	2.52	34.55	*p* < 0.001
Colorectal cancer	1267	4.43	2.01	9.74	*p* < 0.001
Esophageal cancer	214	13.11	5.44	31.62	*p* < 0.001
Hematopoietic tumor	1000	15.80	7.94	31.44	*p* < 0.001
Head and neck cancer	372	9.76	4.19	22.69	*p* < 0.001
Lung cancer	924	10.32	4.99	21.33	*p* < 0.001
Musculoskeletal cancer	75	14.82	3.96	55.48	*p* < 0.001
Stomach cancer	1285	3.34	1.50	7.46	0.003

NS: not significant.

## Data Availability

Data not available due to ethical restrictions.

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
