# Peer review of "Effects of Perioperative Oral Management in Patients with Cancer"

_jcm, 2022, doi:10.3390/jcm11216576_

Round 1

Reviewer 1 Report

Thank you for the opportunity to review this manuscript titled “Effects of perioperative oral management in patients with cancer”. The authors examined the effects of POM on pneumonia prevention and mortality and found that POM may be an effective adjuvant therapy for patients with cancer as mortality was significantly lower in patients who received POM. Overall, this is an interesting topic, clearly of relevance and the manuscript is well-written. I provided some minor comments for the authors to address.

1.     Line 23: The authors should report which the cancer types for which mortality was lower in the POM group.

2.     Line 59: The authors should provide more information about the DPC database for the readers.

3.     Line 96, Table 1: As Table 1 describes the baseline characteristics, the authors should not provide descriptive results for the outcomes pneumonia and death.

Author Response

Point 1: Line 23: The authors should report which the cancer types for which mortality was lower in the POM group.

Response 1:

We greatly appreciate your helpful suggestions.

We added lung and pancreatic cancers with significantly lower mortality rates with POM to the manuscript (Line 23-24).

Point 2: Line 59: The authors should provide more information about the DPC database for the readers.

Response 2:

We apologize for the lack of information. More detailed description of the DPC database and references were added to the methodology (Line 61-62, 67-70).

Point 3: Line 96, Table 1: As Table 1 describes the baseline characteristics, the authors should not provide descriptive results for the outcomes pneumonia and death.

Response 3:

We greatly appreciate your helpful suggestions.

We deleted results for pneumonia and death from the manuscript and Table 1.

Reviewer 2 Report

This paper addresses an important problem in oncologic surgery: the role of perioperative oral management (POM) for the prevention of pneumonia in the surgical setting. The database is impressive, accounting for almost 10,000 cases, and the results suggest that POM may be a valid tool in the perioperative management of certain oncologic patients.

Paradoxically, the main limitation of this study lies in the population size:  the number of patients included in several cancer subgroups (musculoscheletal, skin cancer etc) is just too small to make a real impact and to provide suggestive results.

 However, this study may provide the fundaments for a deeper analysis of the effects of oral management in the postoperative outcomes of patients undergoing oncologic surgery.

Author Response

This paper addresses an important problem in oncologic surgery: the role of perioperative oral management (POM) for the prevention of pneumonia in the surgical setting. The database is impressive, accounting for almost 10,000 cases, and the results suggest that POM may be a valid tool in the perioperative management of certain oncologic patients.Paradoxically, the main limitation of this study lies in the population size:  the number of patients included in several cancer subgroups (musculoscheletal, skin cancer etc) is just too small to make a real impact and to provide suggestive results. However, this study may provide the fundaments for a deeper analysis of the effects of oral management in the postoperative outcomes of patients undergoing oncologic surgery.

Response:

Thank you very much for your review.

In Japan, gastric, colon, lung, and prostate cancers are the most common cancers in men, while breast,colon, stomach, and lung cancers are the most common cancers in women[1].

Therefore, POM is often performed as a target for these cancers. As you pointed out, it is difficult in this study to evaluate the effect of POM on cancers with a small number of cases among the subgroups. We believe that further analysis with additional case numbers is needed in the future.

Since the description of the DPC database was insufficient, a more detailed description was added to the Methodology (Line 61-62, 67-70).

  1. Katanoda, K.; Hori, M.; Saito, E.; Shibata, A.; Ito, Y.; Minami, T.; Ikeda, S.; Suzuki, T.; Matsuda, T. Updated Trends in Cancer in Japan: Incidence in 1985-2015 and Mortality in 1958-2018-A Sign of Decrease in Cancer Incidence. Epidemiol. 2021, 31 (7), 426–450